# Tryptophanyl-tRNA Synthetase as a Potential Therapeutic Target

**DOI:** 10.3390/ijms22094523

**Published:** 2021-04-26

**Authors:** Young Ha Ahn, Se-Chan Oh, Shengtao Zhou, Tae-Don Kim

**Affiliations:** 1Department of Obstetrics and Gynecology, Key Laboratory of Birth Defects and Related Diseases of Women and Children of MOE, State Key Laboratory of Biotherapy, West China Second University Hospital, Sichuan University and Collaborative Innovation Center, Chengdu 610041, China; larica@snu.ac.kr; 2Immunotherapy Convergence Research Center, Korea Research Institute of Bioscience and Biotechnology, Daejeon 34141, Korea; medic305@kribb.re.kr; 3Department of Functional Genomics, KRIBB School of Bioscience, Korea University of Science and Technology, Daejeon 34113, Korea

**Keywords:** tryptophanyl-tRNA synthetase, sepsis, cancer, Alzheimer’s disease, IFN-γ, kynurenine pathway, tryptophan metabolism

## Abstract

Tryptophanyl-tRNA synthetase (WRS) is an essential enzyme that catalyzes the ligation of tryptophan (Trp) to its cognate tRNA^trp^ during translation via aminoacylation. Interestingly, WRS also plays physiopathological roles in diseases including sepsis, cancer, and autoimmune and brain diseases and has potential as a pharmacological target and therapeutic. However, WRS is still generally regarded simply as an enzyme that produces Trp in polypeptides; therefore, studies of the pharmacological effects, therapeutic targets, and mechanisms of action of WRS are still at an emerging stage. This review summarizes the involvement of WRS in human diseases. We hope that this will encourage further investigation into WRS as a potential target for drug development in various pathological states including infection, tumorigenesis, and autoimmune and brain diseases.

## 1. Introduction

Aminoacyl-tRNA synthetases (ARSs) are essential enzymes that ligate amino acids to cognate tRNAs in protein synthesis. The catalytic activities of ARSs play essential roles in maintaining cell viability; however, they are also versatile and multifunctional proteins regulated by diverse control mechanisms [1,2,3,4]. Several features distinguish eukaryotic ARSs from their prokaryotic homologs. The addition of extra domains and sequence adaptations contribute to cellular functions, including cytokine activity associated with inflammation, apoptosis, angiogenesis, and tumorigenesis in mammalian synthetases [5,6]. This review focuses on the roles of eukaryotic tryptophanyl-tRNA synthetase (WRS) in pathological states and its clinical potential as a pharmacological target.

Elevated levels of WRS are expressed in the bovine pancreas, and its various truncated forms are secreted into bovine pancreatic fluid. Thus, an important role of WRS beyond protein translation has long been suspected [7,8,9]. Eukaryotic WRS exists as a free cytosolic enzyme that self-aggregates to form large homo-oligomers, rather than associate with a multi-tRNA synthetase complex (MSC). This property of WRS has been identified in higher eukaryotes, but not in prokaryotes [10,11].

Evolutionarily, ARSs have added new domains that have no apparent connection with aminoacylation. Human WRS has a unique N-terminal extension domain of about 150 amino acids that is not present in its prokaryotic counterpart. The extension domain is composed of a vertebrate-specific extension of about 50 amino acids, also known as WHEP domain, named after initials of (underlined) WRS, histidyl-tRNA synthetase (HRS), and glutamyl-prolyl-tRNA synthetase (EPRS). WHEP domain includes a specific helix-turn-helix motif and is involved with diverse interactions with other proteins [5,6,12,13,14] (Table 1). Elimination of WHEP domain does not markedly inhibit in vitro aminoacylation activities of several synthetases, indicating that the WHEP domains themselves may provide a non-canonical function unrelated to aminoacylation [15,16].

Among 20 ARSs, EPRS and WRS have similarities, in that they have WHEP domains and are regulated by interferon (IFN)-γ. Human EPRS contains three WHEP domains involved in the formation of the IFN-γ activated inhibitor of translation (GAIT) complex, and controlling the translation of vascular endothelial growth factor A (VEGFA) and ceruloplasmin (Cp) [17,18]. Notably, WRS is the only ARS whose expression is induced by IFN-γ [19,20,21,22,23], the action of which is supposedly mediated by WHEP domain. WRS is also rapidly secreted from immune cells in response to both bacterial and viral infections, suggesting a critical role in inflammatory response. In contrast, other ARSs were not secreted from monocytes upon in vitro microbial infections [24,25].

Full-length WRS (FL-WRS) is alternatively spliced or truncated into mini-WRS (residues 48–471). Proteolytic digestion of WRS by extracellular proteases also produces the N-terminally truncated variants T1-WRS (residues 71–471) and T2-WRS (residues 94–471) [20,22,26]. The expression of these truncated variants is stimulated by IFN-γ, which plays a central regulatory role in anti-angiogenesis [27,28,29].

Understanding of the biological functions of truncated WRS variants in vascular homeostasis [26,30,31,32] as well as the structure and properties of secreted WRS has progressed significantly [11,20,21,24,25,33]. Numerous studies have associated WRS with infection, cancer, autoimmunity, and brain diseases. Considering that WRS is secreted during bacterial and viral infections, it has been recently identified as a promising biomarker in patients with sepsis [34]. This review summarizes the physiopathological roles of WRS that have been reported to date and discusses WRS as a potential therapeutic target for human pathologies, especially infection, cancer, autoimmune, Alzheimer’s disease (AD).

## 2. WRS as a Biomarker of Immune Response

### 2.1. WRS Expression in Systemic Inflammatory Response

Defining protein expression profiles in various cell types in tissues and organs is critical for understanding their unique physiological characteristics in humans. Addressing this has a major impact not only on efforts to identify target protein functions, but also for applied medical research into disease management, such as pharmaceutical drug development and biomarker discovery [45,46].

RNA and protein expression data generated by the Human Protein Atlas project indicates that WRS is abundantly expressed in monocytes (http://www.proteinatlas.org/ENSG00000140105-WRS/blood, accessed on 17 December 2020). Monocytes are early responders to pathogens and maintain vascular homeostasis in acute infections [47]. Like proteins that specifically act on monocytes, WRS is abundantly expressed during differentiation from monocytes to monocyte-derived macrophages (MDMs) or dendritic cells (DCs) [48,49]. In addition, microbial infection and IFN-γ stimulation causes monocytes, but not B, T, or natural killer (NK) cells, to produce and secrete WRS [19,24,25]. Classical (CD14^+^CD16^−^), intermediate (CD14^+^CD16^+^), and non-classical (CD14^−^CD16^+^) monocytes have been characterized and relative ratios (%) of monocyte subsets have been determined to aid in understanding the pathogenesis of infectious and other inflammatory disorders. WRS is more heavily expressed in intermediate and non-classical monocytes compared to the classical subset (Figure 1A). At the early stages of infection, FL-WRS is secreted from monocytes and directly binds to TLR4-MD2 complexes on macrophages to activate phagocytosis [24]. Initial FL-WRS secretion is IFN-γ-independent and acts as a warning to prime innate immunity. Similar to these pro-inflammatory functions of FL-WRS, classical monocytes also play a crucial role in phagocytosis during the initial inflammatory response [50].

Non-classical monocytes have been widely viewed as being anti-inflammatory in order to maintain vascular homeostasis. Intermediate and non-classical monocytes gradually increase and expand in infectious diseases such as sepsis, in which they constitute ~50% of all monocytes [47,51,52]. Pathological conditions such as sepsis and septic shock are characterized by the elevated expression of proteases such as fibrin, neutrophil elastase (NE), and matrix metalloproteinases (MMPs), which simultaneously produce the mini-WRS, T1-WRS, and T2-WRS truncated variants [22,26,53,54]. Anti-inflammatory pathways are concurrently activated at this time, leading to the release of anti-inflammatory cytokines that dampen and, ultimately, terminate the inflammatory response. With the gradual increase in intermediate and non-classical monocytes, a secondary increase in WRS expression is supposed to be occurred to maintain homeostasis (Figure 1B). As the truncated variants have anti-inflammatory and antiangiogenic functions [20,26,30], it is critical to distinguish them from FL-WRS and to evaluate their kinetics during the inflammatory phase. Furthermore, understanding the pathological roles of the truncated variants in systemic inflammatory diseases might contribute to their potential as a new therapeutic target.

### 2.2. WRS as a Prognostic Biomarker in Sepsis

Sepsis is characterized by an overwhelming systemic inflammatory reaction to microbial infection that can lead to severe sepsis and septic shock [55]. Early recognition of sepsis is critical for timely and effective intervention [56]. However, biomarkers that reflect the severity of infection in patients with sepsis have not yet been identified. Although C-reactive protein (CRP) is a traditional biomarker that is elevated in inflammatory states, it has low specificity for diagnosing sepsis [57].

Consistent with previous findings of abundant WRS secretion upon microbial infection in vitro and in vivo [24,25], high levels of WRS are detected in the serum of critically ill patients with sepsis. Choi et al. recently reported that WRS has clinical value for detecting sepsis and predicting mortality among critically ill patients with sepsis. The areas under the receiver operating characteristic curves (AUROCs) for sepsis discrimination with WRS, procalcitonin (PCT), CRP, and IL-6 are 0.864, 0.727, 0.625, and 0.651, respectively [34], indicating that WRS has better predictive value than other clinical factors of sepsis (Figure 2A).

Moreover, existing diagnostic markers of sepsis were only able to detect bacterial sepsis. Among several markers of inflammation and sepsis, PCT is used to identify bacterial infections, but also has the disadvantage of not identifying viral infections. [58,59]. Besides higher sensitivity towards sepsis discrimination of WRS, another advantage of WRS is that it can be used to diagnose sepsis regardless of whether the causal infection is bacterial, fungal, or viral.

Furthermore, biomarkers are needed that can accurately predict mortality due to severe sepsis and septic shock. PCT and CRP levels do not provide statistically significant prediction of 28-day mortality in sepsis patients; however, WRS is significantly associated with 28-day overall mortality among patients with sepsis in intensive care units [34]. Based on these clinical results, WRS secretion at early infection stages is crucial in the recognition and phagocytosis of microbes, which are the main roles of classical monocytes. Sepsis gradually worsens and WRS, which is synthesized *de novo* and secreted upon IFN-γ activation, is likely to function similarly to non-classical monocytes that are involved in pathophysiology associated with anti-inflammatory response and blood vessel remodeling. Secreted WRS is a sensitive and accurate biomarker not only for discriminating sepsis but also for predicting mortality among critically ill patients.

## 3. WRS as a Therapeutic Target in Cancer

### 3.1. WRS as a Target for Anti-Angiogenic Therapy

Several ARSs have been identified as secretory cytokines that control angiogenesis and immune responses in the tumor microenvironment. Fragments of the closely related mammalian tyrosyl-tRNA synthetase (YRS) and WRS are known to regulate angiogenesis [60]. Native FL-WRS does not affect angiogenic signaling; however, mini-WRS, T1-WRS, and T2-WRS, are all angiostatic factors [26,30,32,60,61]. Expression of the truncated WRS variants, like that of many angiostatic factors, is induced by IFN-γ. In particular, T2-WRS has proven potently antiangiogenic in several cell-based assays of vascular endothelial growth factor (VEGF)-induced Matrigel angiogenesis in vitro and in vivo. T2-WRS is a potent inhibitor of retinal angiogenesis in neonatal mice, where it localizes to retinal blood vessels [30]. Moreover, the inhibitory activity of T2-WRS in endothelial cells (ECs) in vitro abrogates cellular responses to flow-induced fluid shear stress, including endothelial nitric-oxide synthase (eNOS), extracellular signal-regulated kinase (ERK1/2), and Akt activation [31]. The eight-residue D382-TIEEHR-Q389 sequence, which is located within the tRNA anticodon-binding (TAB) domain, plays a crucial role in the angiostatic activity of the truncated variants. This implies that WRS uses the same domain for both antiangiogenic receptor binding and classical protein synthesis [2,61].

Vascular endothelial (VE)-cadherin is a calcium-dependent adhesion molecule that is selectively expressed at EC intercellular junctions and is essential for normal vascular development. T2-WRS binds at EC intercellular junctions and VE-cadherin has been identified as a receptor for T2-WRS. The binding of T2-WRS to human umbilical vein ECs (HUVECs) inhibits VEGF-induced ERK activation and EC migration [32]. Adam et al. suggested that human WRS manifests antiangiogenic activity only when a eukaryote-specific N-terminal extension is removed, due to steric hindrance that blocks Trp2 and Trp4 of VE-cadherin from accessing the active site pocket [62]. T2-WRS contains a Rossmann fold nucleotide-binding domain that is conserved throughout all prokaryotic and eukaryotic WRSs. FL-WRS, mini-WRS, and T1-WRS variants retain aminoacylation activity, whereas T2-WRS does not [30]. Thus, T2-WRS may apparently be produced only to inhibit angiogenesis without contributing to aminoacylation (Figure 2B and Table 2). Proteases including fibrin, NE, and MMPs are abundant in tumor microenvironments or at sites of infection, where they produce T2-WRS by cleaving the N-terminus. This could explain why WRS secreted by IFN-γ activation is truncated to T2-WRS, and its role in inhibiting angiogenesis is of great importance in clinical approaches targeting WRS.

### 3.2. Implications of Angiostaic WRS in Cancer Metastasis

ARS expression profiles can be a useful prognostic tool for cancers, as they correlate with overall patient survival in each cancer type. WRS is dysregulated in different cancers, including ovarian, cervical, colorectal, oral squamous cell carcinoma (OSCC), uveal melanoma (UM) and pancreatic cancers [63,64,65,66,67,68]. In ovarian cancer, WRS protein levels were up-regulated in highly malignant clear cell adenocarcinoma when compared with mucinous ovarian adenocarcinoma [63]. The expression level of WRS is increased in cervical carcinoma tissue when compared with normal tissue [64]. WRS is overexpressed in OSCC tissues when compared with adjacent normal tissues. Importantly, WRS levels are significantly higher in tumor cells from metastatic lymph nodes than in primary tumor sites [65,69]. Bioinformatics analysis based on Gene Expression Omnibus (GEO) database showed that WRS expression in UM metastatic cancer was significantly higher than that in non-metastatic group. Kaplan-Meier analysis based on The Cancer Genome Atlas (TCGA) database showed that high WRS expression was associated with lower survival [66]. Most cancer-associated mortality is due to tumor cell metastasis to other organs. WRS is expressed abundantly in patients who did not relapse after surgery for triple-negative breast cancer (TNBC), when compared with those who did [70]. Moreover, low WRS expression is associated with an increased risk of lymph node metastasis and reduced survival among patients with colon cancer [68].

Tumor cells metastasize via lymphatic or hematogenous dissemination. Although lymphatic metastasis can occur via extant vessels, evidence supports the notion that metastasis is significantly improved when lymphatic vessel density increases due to lymphangiogenesis. For example, VEGF-C stimulates the growth of lymphatic endothelium and is overexpressed in breast cancer cells. Stimulation by VEGF-A and -C is also associated with lymphatic vessel formation and sentinel node metastasis [71,72,73]. WRS are overexpressed at stages of cancer metastasis and T2-WRS has a potent inhibitory effect against VEGF-induced vessel formation. However, whether the angiostatic activity of T2-WRS is associated with lymphangiogenesis during metastatic progression remains unknown.

The truncated variants are antagonistic in the control of angiogenesis and immune stimulation and has potential as a therapeutic target for cancer. In contrast, FL-WRS shows the opposite function compared to the truncated variants, suggesting that caution is needed when interpreting the relationship between FL-WRS and its truncated variants. Furthermore, the targets of each truncated WRS variant need to be validated and their effectiveness as an immuno-oncology target should be confirmed.

## 4. Pathological Role of WRS in Alzheimer’s Disease

Alzheimer’s disease (AD) is an age-dependent neurodegenerative disorder and the most common cause of dementia [74]. It is characterized by neurofibrillary tangles (NFTs) composed of paired helical filaments and extracellular senile plaques containing aggregated amyloid fibrils and non-amyloid components in the brain [71]. Neither an effective treatment nor a biochemical drug for AD is currently available.

Neuronal cell death triggers the release of cytoplasmic proteins, some of which interact with Amyloid-β (Aβ). Specifically, glyceraldehyde-3-phosphate dehydrogenase (GAPDH) has been shown to interact with neurodegenerative disease-related proteins including β-amyloid precursor protein (AβPP). Aβ induces disulfide bond formation and aggregation of GAPDH, possibly due to the formation of neurotoxic aggregates in AD [75,76,77,78]. The amount of GAPDH disulfide bonding is increased in detergent-insoluble extracts from patients with AD when compared with age-matched controls [79]. Interestingly, bovine WRS forms complexes with GAPDH. This complex formation between WRS and GAPDH i) does not influence aminoacylation activity and ii) predominantly involves the dispensable N-terminal domain of WRS [72]. The tendency of mammalian WRS to self-aggregate has been observed by electron microscopy, small-angle X-ray scattering, and biochemical experiments [10,73,74]. Oligomeric WRS is found in cytotoxic fibrils and extracellular plaque-like WRS aggregates detected in AD have features similar to extracellular senile plaques and NFTs (Figure 2C). Electron microscopy has shown intensive fibril formation with a synthetic N-terminal peptide of WRS, corresponding to residues Ser32 to Tyr50, whereas fewer and less organized fibrils were formed by the C-terminal peptide (Glu414 to Val437) [11].

Tryptamine is an inhibitor that competes with Trp for binding to the WRS active site [75]. The expression of WRS is decreased in the cytoplasm and elevated in the detergent-insoluble “cytoskeleton” fraction of tryptamine-treated human neuronal cells. Moreover, about three-fold higher WRS immunogold reactivity was associated with NFTs than with the cytoplasm of tryptamine-treated human neuronal cells [74]. Paley et al. reported in tryptamine-treated cells that WRS is secreted into the extracellular space as either a free protein or within vesicles extending from the cytoplasm and then pinched off from the plasma membrane. Extracellular vesicles fuse in congophilic WRS^+^ plaques in tryptamine-treated cultures and in the brain in AD. Prominent WRS immunoreactivity is associated with plasma and the vesicle membranes of satellites and degenerated neurons in the brain with AD [76]. Biochemically purified bovine WRS is highly susceptible to aggregation [10] and recombinant human WRS and N-terminal synthetic peptides self-assemble in fibrils [11]. These findings provide a new perspective on the versatility of WRS functions and indicate that WRS is involved in the pathology of AD. Therefore, further studies are needed to clarify the relationship between human WRS and AD and to investigate whether WRS could be a novel therapeutic target, in particular by focusing on anti-WRS therapy to prevent or hinder the clinical progression of AD.

## 5. Immunological Role of WRS in Trp Metabolism

### 5.1. Increased Trp Production by WRS in Cancer Cell

Tryptophan (Trp) is an essential amino acid that is important for cell survival and proliferation. Indoleamin-2,3-dioxygenase (IDO) is the initial and rate-limiting enzyme for Trp degradation through the kynurenine pathway (KP). Trp catabolism suppresses antitumor immune responses in pathological states such as cancer [80,81]. IFN-γ enhances expression of IDO as well as WRS. Both IDO and WRS are involved in regulating the immune response by modulating Trp metabolism. IFN-γ induces WRS to increase Trp uptake into cells through a process that has high affinity and selectivity for Trp [82]. Consistent with this, Adam et al. reported that Trp depletion mediated by IDO1 and tryptophan-2,3-dioxygenase (TDO2) upregulates WRS expression to increase Trp production in cancer cells. Due to WRS mediated-Trp synthesis, cancer cells are thought to maintain protein synthesis and proliferate despite low Trp levels [83]. The depletion of Trp induced by IDO up-regulation results in specifically inhibited T cell proliferation, thus facilitating tumor escape from immune surveillance. Additionally, increased WRS expression and WRS-mediated Trp production directly facilitate cancer cell proliferation and survival, implying that both IDO and WRS are associated with immune evasion by cancer cells. A new approach to controlling Trp production by WRS might therefore resolve the redundancy of Trp metabolites in cancer immunotherapy. However, rather than pathological role of WRS acting on cancer itself, the involvement and regulation of WRS in tolerogenic immune response and autoinflammatory disease are prominent topics of interest in the context of therapeutics.

### 5.2. Implications of Increased WRS in Tolerogenic Immune Response

Under healthy conditions, Trp-degrading activity is abundant in the lungs, intestines, and particularly in placental tissues [84]. Notably, WRS transcripts are also abundantly expressed in placenta and lung tissues, mostly known as immune privileged sites (https://www.proteinatlas.org/ENSG00000140105-WARS/tissue, accessed on 17 December 2020). The importance of Trp depletion in the placenta is emphasized in maintaining immune privilege to create an environment that suppresses T cell activity and defends itself against rejection [85,86]. The expression of IDO1 and TDO2 is positively correlated with WRS, which in turn is significantly correlated with the expression of T cell markers. Recombinant soluble cytotoxic T lymphocyte antigen-4 (CTLA-4-Fc) is clinically effective for suppressing T cell activation in autoimmune diseases such as rheumatoid arthritis (RA). The expression of IDO and WRS in immune cells, such as DCs and CD4^+^ T cells is increased by CTLA-4-Fc. Notably, CD8^+^ T cells from CTLA-4-Fc-treated peripheral blood mononuclear cells (PBMCs) express increased levels of WRS but not IDO, suggesting that WRS-mediated regulation and IDO are involved in the immune tolerance mechanism of CTLA-4 [87]. Overexpression of WRS in T cells from patients with RA is apparently related to the pathogenesis of this disease. Although IDO is overexpressed in DCs in the synovial joints of patients with RA, IDO^+^ DCs do not have sufficient immunosuppressive ability when WRS is overexpressed in T cells [88,89,90]. The activity of IDO is decreased, whereas WRS expression is increased in T cells from patients with immune thrombocytopenia (ITP) [91] and Graves’ disease [92], which are autoimmune disorders that result in platelet destruction and hyperthyroidism, respectively.

The question remains as to whether WRS, which is specifically overexpressed in T cells in patients with autoimmune diseases, indirectly affects Trp metabolism or whether WRS is a cause of autoimmune diseases as a result of its direct involvement in the immune tolerance of T cells. Ongoing clinical trials have shown that IDO inhibitors are well tolerated in cancer, systemic diseases, and central nervous system disorders targeting the pathogenic KP. Nonetheless, results using monotherapy have been somewhat limited and need an additional emerging target to overcome IDO redundancy. Increased WRS expression is likely to act as a pathological factor, not only in the tumor microenvironment, but also in Trp metabolism in autoimmune diseases such as RA. The identification of WRS as a biomarker and the development of selective WRS inhibitors might help improve therapeutic outcomes when combined with IDO inhibitors.

## 6. Conclusions and Perspectives

The various features that distinguish eukaryotic WRS from prokaryotic WRS or other ARSs suggest that WRS has evolutionarily acquired diverse functions that are closely associated with human disease. Human WRS (hWRS) shows great potential as a suitable drug target, having non-canonical functions that underlie its physiopathological roles in human diseases, including infection, cancer, and neurodegenerative diseases (Figure 2). Upon infection, hWRS is promptly secreted from monocytes for the priming of innate immunity [24]. Secretory hFL-WRS acted as a therapeutic immune-stimulatory agent and showed a promising potential as a novel biomarker for diagnosis of sepsis [34]. Moreover, alternatively spliced and truncated variants, including mini-WRS, T1-WRS, and T2-WRS, exhibit anti-angiogenic and anti-inflammatory properties. Each truncated WRS variant is likely to exert its own specific biochemical effect for the maintenance of homeostasis in pathological states. hWRS, together with oligomerized plaques, contributed to the pathogenesis of Alzheimer’s disease.

The hWRS research field has extended from basic research to antibody and diagnostic development. However, the development of hWRS-targeted therapeutics remains a challenge. Above all, sensitive analytical methods are required to distinguish each WRS variant, owing to their different mechanism of action and pathological roles. Subsequently, the therapeutic effects acting on the hWRS target sites can be easily and accurately monitored. Achieving this goal will lead a progressive development in the field of hWRS-targeted therapeutics. Currently, JW Bioscience, a company, is actively developing WRS diagnostic kits for sepsis and the products are undergoing clinical trials. Although significant progress has been made in the research on hWRS as a therapeutic target, a qualitative study is required to determine the unknown features of ARS variants in various diseases.

## Figures and Tables

**Figure 1 ijms-22-04523-f001:**
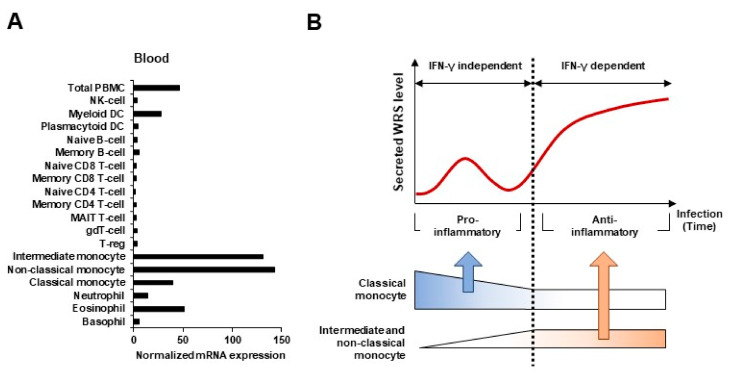
WRS as a biomarker of immune response during infection. (**A**) Expression levels of WRS transcripts in various blood cell types. The expression of WRS is enhanced in intermediate and non-classical monocytes. (**B**) Kinetics of WRS and three monocyte subsets (classical, intermediate, and non-classical) in inflammatory responses caused by infection. The secretion of WRS increased consistently with an increase in the numbers of intermediate and non-classical monocytes.

**Figure 2 ijms-22-04523-f002:**
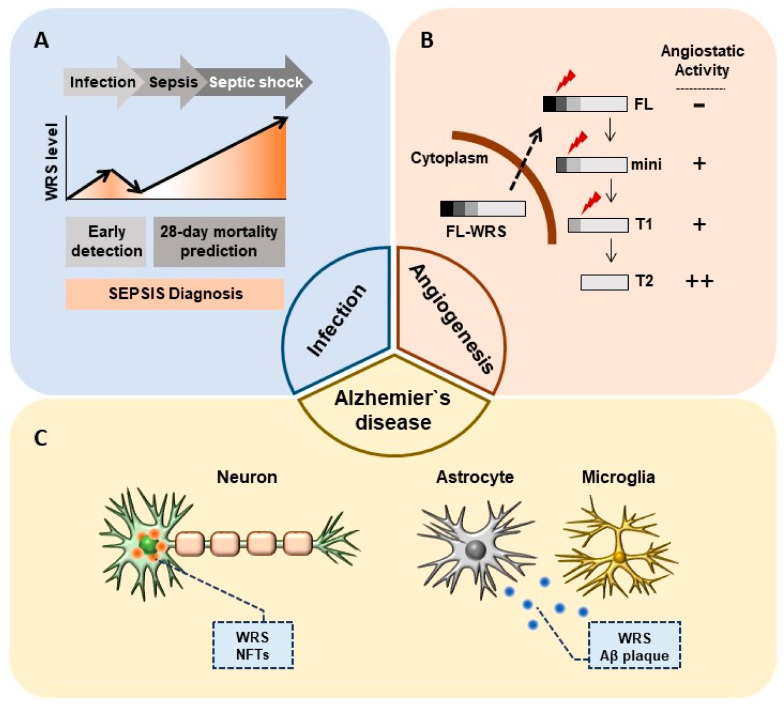
Non-canonical physiopathological roles of WRS. (**A**) Secreted WRS as a sepsis biomarker. Secreted WRS is a potential biomarker not only for the early detection of sepsis but also for predicting 28-day mortality. (**B**) Different angiostatic activities of each WRS variant in angiogenesis. Mini-, T1-, and T2-WRS have antiangiogenic effects that may be related to metastasis in tumor microenvironments. (**C**) Intracellular oligomerized WRS with NFTs and extracellular plaque-like WRS, along with Aβ plaques, are detected in AD.

**Table 1 ijms-22-04523-t001:** Schematic structure of ARSs containing WHEP domain. EPRS and MRS are components of the multi-tRNA synthetase complex (MSC) and the cell regulatory activities are controlled by specific phosphorylation [5,35,36,37,38,39]. WRS, HRS, and GRS exist in a free form and are induced under various pathological conditions [12,40,41,42,43,44]. The regulatory activities of WRS are controlled by proteolytic cleavage. The secreted WRS are cleaved and produce WRS variants such mini-, T1- and T2-WRS. GST, Glutathione S-transferase; CD, catalytic domain; ERS, glutamyl-tRNA synthetase; PRS, prolyl-tRNA synthetase; EPRS, glutamyl-prolyl-tRNA synthetase; MRS, methionyl-tRNS synthetase; HRS, histidyl-tRNA synthetase; GRS, glycyl-tRNA synthetase.

	ARSs	Schematic Structure	Alternative Variants	Proteolytic Cleavage	References
MCS form	EPRS	1 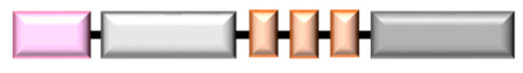 1512	EPRS^N1^		[17,18,35,36]
MRS	1 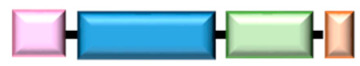 900			[37,38,39]
Free form	WRS	1 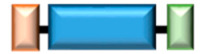 471	mini-WRS	T1-WRST2-WRS	[6,13,20,22,27,28,30,31,32]
HRS	1 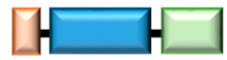 509	HRSΔC		[40,41,42]
GRS	1 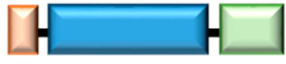 739			[43,44]

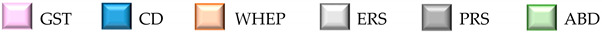

**Table 2 ijms-22-04523-t002:** Schematic structure and angiostatic activity of human WRS variants. The Rossmann fold (RF) and anticodon binding domain (ABD) are well conserved in all WRS variants. The eukaryotic-specific extension (ESE) is common in eukaryotic WRS. Human full-length (FL)-WRS has a vertebrate-specific extension (VSE), also known as WHEP domain. The aminoacylation or angiostatic activities of WRS variants are indicated in +/−.

WRS Variants	Schematic Structure	Aminoacylation Activity	Angiostatic Activity
FL-WRS	1 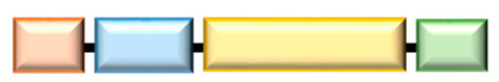 471	+	−
mini-WRS	48 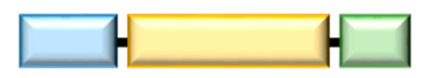 471	+	+
T1-WRS	71 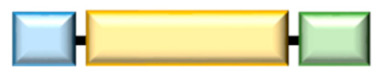 471	+	+
T2-WRS	94 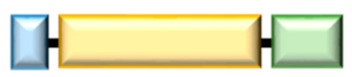 471	−	++

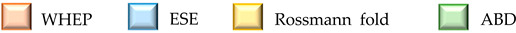

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
