# Peer review of "Tryptophanyl-tRNA Synthetase as a Potential Therapeutic Target"

_ijms, 2021, doi:10.3390/ijms22094523_

Round 1

Reviewer 1 Report

The authors present a comprehensive overview of non-canonical functions of WRS and stress its potential as a therapeutic target in disease. Although the topic is interesting and relevant, the review needs some restructuring to make it more understandable for the non-specialist reader.

I am not an expert in these non-secreted roles of WARS1 and therefore cannot judge whether the adequate citations have always been used, and whether all interpretations of the cited literature are always correct. 

Major comments: 

  • Many sentences/statements need appropriate referencing, in particular in the introduction: Page 1 line 42-43; p.2 line 6; p.2 line 10-11; p.2 line 12-13; p.2 line 24-25, p.4 line 45-47 are just a few examples that I noticed.
  • The review would benefit from a table or figure illustrating the various truncated variants of WRS and the reported roles of each, with references (maybe best in a table?). 
  • The figures should be referred to in the text where appropriate (which does not happen now). 
  • P4, section 3: a major part here concerns roles of WRS in angiogenesis, independent of the tumour context. The review would benefit from a separate section about WRS in the context of development (which can be focused on angiogenesis). 
  • P4. lines 41 and below: This is very difficult to understand for the non-expert reader, and I am not sure whether the way metastatic/non-metastatic modules are being referred to are correct. This needs rewriting and appropriate referencing. 
  • p.5 section 4: As far as I understand, this mostly concerns enzymatic roles of WRS. It is not clear to me why this is being discussed here. If this is thought appropriate, it needs to be introduced. In addition, explanation about how WRS modulates Trp synthesis and catabolism is required because the structure of the section is confusing. 
  • p.7 section 6: I am not sure why Met or Tyr ARS for example are not mentioned here as playing a role in hereditary neuropathies? Similar to the previous section, I am not sure whether this section is entirely relevant in this review, as it is related to the enzymatic functions of these ARS's (as far as I am aware?). If it is deemed relevant, it should be introduced as such. In addition, it might be interesting to distinguish mitochondrial from cytoplasmic enzymes, and also to refer to homozygous/compound heterozygous mutations (or the fact that these have not been observed for WARS1?). 

Minor comments: 

  • Is there a reason to authors use WRS, rather than WARS or WARS1/2? 
  • The abstract needs re-writing, to make clear that the authors focus on non-enzymatic/secreted roles of WRS (if that's really the case?)
  • p. 1 line 37-38: it is not clear to me whether it is only the secreted WRS that exists as a free cytosolic enzyme? 
  • p. 3 lines 25-30: I don't think the discussion of PCT is relevant here. 
  • p. 6 line 20: maybe re-explain the abbreviation AD here? 
  • p. 6 lines 25-28: GAPDH seems to come out of nowhere here. Maybe it would be better to first introduce the aggregation of WRS, then mention that it can complex with GAPDH, and then explain thoughts about GAPDH aggregation (if it is deemed necessary). 

Author Response

Reviewer #1:

The authors present a comprehensive overview of non-canonical functions of WRS and stress its potential as a therapeutic target in disease. Although the topic is interesting and relevant, the review needs some restructuring to make it more understandable for the non-specialist reader.

I am not an expert in these non-secreted roles of WARS1 and therefore cannot judge whether the adequate citations have always been used, and whether all interpretations of the cited literature are always correct. 

Major comments: 

  • Many sentences/statements need appropriate referencing, in particular in the introduction: Page 1 line 42-43; p.2 line 6; p.2 line 10-11; p.2 line 12-13; p.2 line 24-25, p.4 line 45-47 are just a few examples that I noticed.
  • Answer: We appreciate the reviewer for the in-depth understanding of this work and kind comments. We presented the appropriate references in Page 2 line 5, 10-11, 14, 19, 21-22, 37; p.5 line 12, 43; p.7 line 8, 44-45
  • The review would benefit from a table or figure illustrating the various truncated variants of WRS and the reported roles of each, with references (maybe best in a table?). 
  • Answer: According to reviewer’s comment, we illustrated the schematic structure and roles of various WRS variants in Table 1 and Table 2.
  • The figures should be referred to in the text where appropriate (which does not happen now). 
  • Answer: We referred the figures and tables in the text. (p.2 line 3, 47; p.3 line 13; p.4 line 5; p.5 line2; p.6 line 10; p.7 line 42)
  • P4, section 3: a major part here concerns roles of WRS in angiogenesis, independent of the tumour context. The review would benefit from a separate section about WRS in the context of development (which can be focused on angiogenesis). 
  • Answer: We separated this section into 3.1. WRS as a target for anti-angiogenic therapy and 3.2. Implications of angiostatic WRS in cancer metastasis.
  • P4. lines 41 and below: This is very difficult to understand for the non-expert reader, and I am not sure whether the way metastatic/non-metastatic modules are being referred to are correct. This needs rewriting and appropriate referencing. 
  • Answer: We misunderstood the paper “Disease biomarker identification from gene network modules for metastasized breast cancer’. We also couldn’t understand the meaning of metastatic/non-metastatic module in that paper, so deleted all the mentions about the paper. We added other appropriate references and re-wrote the text (p.5 line 10-26)
  • p.5 section 4: As far as I understand, this mostly concerns enzymatic roles of WRS. It is not clear to me why this is being discussed here. If this is thought appropriate, it needs to be introduced. In addition, explanation about how WRS modulates Trp synthesis and catabolism is required because the structure of the section is confusing.  In section 5, rather than pathological role of WRS acting on cancer itself, we are more interested in the functional role of WRS in immune cells. Therefore, we suggested that researches on the involvement and regulation of WRS in tolerogenic immune response and autoinflammtory disease should be further studied in a view of therapeutics in section 5.2.  
  •  
  • Answer: We understood the reviewer’s concern. For better understanding, we separated this section into 5.1. Increased Trp production by WRS in cancer and 5.2. Implications of increased WRS in tolerogenic immune response.
  • p.7 section 6: I am not sure why Met or Tyr ARS for example are not mentioned here as playing a role in hereditary neuropathies? Similar to the previous section, I am not sure whether this section is entirely relevant in this review, as it is related to the enzymatic functions of these ARS's (as far as I am aware?). If it is deemed relevant, it should be introduced as such. In addition, it might be interesting to distinguish mitochondrial from cytoplasmic enzymes, and also to refer to homozygous/compound heterozygous mutations (or the fact that these have not been observed for WARS1?). Minor comments: 
  • Answer: We understood the reviewer’s concern. In fact, almost all ARSs’ mutations are related with neuropathy diseases such as CMT or HLD. In this review, there is no significant impact to mention mutations in ARSs as a WRS therapeutic target. Therefore, according to reviewer’s comment, we excluded this section “WRS mutations in inherited neuropathic diseases”.
  • Is there a reason to authors use WRS, rather than WARS or WARS1/2? 
  • Answer: Mostly, cytoplasmic tryptophanyl-tRNA synthetase is expressed as TrpRS, WARS, WARS1, and WRS. This nomenclature is used similar to the other ARSs. In the early 2000s, research on the complexity of ARSs became active, revealing the relationship of each ARSs and used the most abbreviated form such as WRS, MRS, EPRS, etc. After then, it can be often seen those expressions of abbreviation in many papers.
  • The abstract needs re-writing, to make clear that the authors focus on non-enzymatic/secreted roles of WRS (if that's really the case?)
  • Answer: Although secretory WRS is more attractive as drug target, it is not necessary to exclude intracellular WRS from the possibility of drug targeting. Therefore, we didn’t modify the abstract, except for deletion “secretory” in page 1, line 19.
  • p. 1 line 37-38: it is not clear to me whether it is only the secreted WRS that exists as a free cytosolic enzyme? 
  • Answer: We corrected “Secreted eukaryotic WRS” to “Eukaryotic WRS”. (page 1, line 37).
  • p. 3 lines 25-30: I don't think the discussion of PCT is relevant here. 
  • Answer: We understood the reviewer’s concern. According to reviewer’s comment, we modified the statement about PCT (page 4, line 6)
  • p. 6 line 20: maybe re-explain the abbreviation AD here? 
  • Answer: According to reviewer’s comment, we re-explained the abbreviation AD (page 5, line 43)
  • p. 6 lines 25-28: GAPDH seems to come out of nowhere here. Maybe it would be better to first introduce the aggregation of WRS, then mention that it can complex with GAPDH, and then explain thoughts about GAPDH aggregation (if it is deemed necessary). 

Answer: We understood the reviewer’s concern. There are few reports on the effect of the interaction of GAPDH and WRS on AD. However, we thought the finding that the interaction of WRS with GAPDH probably have a significance. In addition, we wrote statement about the pathological relevance of GAPDH in AD (page 5, line 48)  

Reviewer 2 Report

Aminoacyl-tRNA synthetases are key enzymes implicated in mRNA translation. In the present article, the authors review some of the implication of the tryptophanyl-tRNA synthetase (WRS) into various human diseases. It thus illustrates the variability of functions that can perform the WRS protein, beyond its canonical role in protein biosynthesis through aminoacylation of cognate tRNATrp. The review is well written and of interest for a large audience. The following minor comments should however be considered before submission.

The authors indicate that "WRS is the only ARS whose expression is induced by interferon (IFN)-gamma". They are missing the implication of (IFN)-gamma on EPRS.

Does WRS is the only sepsis biomarkers, or other ARS have been shown to play such a role as well?

In Figure 2 panel B, it would be interesting to add a column (as for the angiostatic activity) with the indication the aminoacylation capability of the different WRS variants.

Figure 2 panel C: what IDO1 and TDO2 stand for should be indicated in the legend of the figure.

Author Response

Reviewer #2:

Comments and Suggestions for Authors

Aminoacyl-tRNA synthetases are key enzymes implicated in mRNA translation. In the present article, the authors review some of the implication of the tryptophanyl-tRNA synthetase (WRS) into various human diseases. It thus illustrates the variability of functions that can perform the WRS protein, beyond its canonical role in protein biosynthesis through aminoacylation of cognate tRNATrp. The review is well written and of interest for a large audience. The following minor comments should however be considered before submission.

The authors indicate that "WRS is the only ARS whose expression is induced by interferon (IFN)-gamma". They are missing the implication of (IFN)-gamma on EPRS.

Answer: We appreciate the reviewer for the kind understanding of this work. According to reviewer’s comment, we modified our manuscript to address missing points about IFN-gamma activated EPRS. (page1, line 40 ~ page2, line 14)

Does WRS is the only sepsis biomarkers, or other ARS have been shown to play such a role as well?

Answer: Systemic inflammation related- secretory ARSs, such as AIMP1, KRS and GRS, also detected in sepsis patients. However, other ARSs are not good biomarkers for microbial infection, showing low ROC for sepsis discrimination. In addition, other ARSs were not secreted from monocytes upon in vitro microbial infections (page 2, line 13).

In Figure 2 panel B, it would be interesting to add a column (as for the angiostatic activity) with the indication the aminoacylation capability of the different WRS variants.

 Answer: According to reviewer’s comment, we illustrated the schematic structure and roles of various WRS variants in Table 1 and Table 2.

Figure 2 panel C: what IDO1 and TDO2 stand for should be indicated in the legend of the figure.

 Answer: To highlight the secretory WRS aspects, the intracellular WRS part was deleted from Figure 2.

Round 2

Reviewer 1 Report

I have no additional comments.